# Diverse Approaches to Insect Control: Utilizing *Brassica carinata* (A.) Braun and *Camelina sativa* (L.) Crantz Oil as Modern Bioinsecticides

**Katarzyna Rzyska** [1,*] **, Kinga Stuper-Szablewska** [1] **and Danuta Kurasiak-Popowska** [2]

[1] Department of Chemistry, Faculty of Forestry and Wood Technology, Poznan University of Life Sciences, ul. Wojska Polskiego 75, 60-628 Poznan, Poland; kinga.stuper@up.poznan.pl

[2] Department of Genetics and Plant Breeding, Faculty of Agronomy, Horticulture and Bioengineering, Poznan University of Life Sciences, ul. Dojazd 11, 60-632 Poznan, Poland; danuta.kurasiak-popowska@up.poznan.pl

* Correspondence: katarzyna.rzyska@up.poznan.pl

**Abstract:** The forest environment is exposed to a number of harmful factors that significantly reduce the resistance of forest stands, often leading to their extinction. In addition to abiotic and anthropogenic factors, biotic factors pose a significant threat to forests, among which insect pests are at the top of the list. Until now, the use of chemical insecticides has been considered the most effective method of pest control, resulting in pesticide residue in the environment. In an effort to minimize the harmful effects of insecticides, the European Union (EU), through EU Commission Implementing Regulations 2022/94, 2021/2081, 2021/795, and 2020/1643, has decided to withdraw from use a number of preparations containing compounds such as phosmet, indoxacarb, alpha-cypermethrin, and imidacloprid, among others. Botanical insecticides appear to be a promising alternative. Among them, plant oils and essential oils have become an innovative solution for controlling pests not only of forests but also of agricultural crops. The purpose of this literature review was to select oilseed plants with great biological potential. The rich chemical compositions of the seeds of *Brassica carinata* (A.) Braun and *Camelina sativa* (L.) Cranz predispose them to use as raw materials for the production of biopesticides with broad mechanisms of action. On the one hand, the oil will provide a physical action of covering pests feeding on a plant with a thin film, which will consequently lead to a reduction in gas exchange processes. On the other hand, the bioactive compounds in it or extracts of fat-insoluble compounds suspended in it and derived from the pomace fraction may have deterrent or lethal effects. This paper presents evidence of the potential action of these raw materials. Preparations based on these oils will not pose a threat to living beings and will not negatively affect the environment, thus allowing them to gain social acceptance.

**Keywords:** bioinsecticides; *Brassica carinata*; *Camelina sativa*; insecticides; plant oils

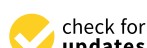

## 1. Introduction

The forest environment is vulnerable to numerous harmful factors that can significantly reduce the resistance of forest stands, often leading to their demise. In addition to abiotic and anthropogenic factors, biotic factors pose significant threats to forests, with insect pests being the most prominent [1]. Pests that occur periodically in large numbers, with such aggregations being known as gradient pest outbreaks, are particularly harmful to the sustainability of forest stands. *Lymantria monacha*, *Dendrolimus pini*, and *Acantholyda posticalis* are the most common pests that can cause damage covering hundreds or even thousands of hectares of forests [2]. These species, during periods of their increased occurrence, can lead to the complete die-off of large areas of stands. It is important to note the severity of the damage that these pests can cause.

There are many methods that can be used to reduce populations of insect pests, but insecticides are definitely the best way to control them [3,4]. Preparations containing

sulfur, arsenic, or mercury, as well those of biological origin, have been used since ancient times [5,6]. The oldest botanical insecticides are pyrethrins extracted from the flowers of the Dalmatian sea-rose (*Tanacetum cinerariifolium*) [7], rotenone from the root of poison deris (*Derris elliptica*) [8], nicotine from tobacco leaves (*Nicotiana tabacum*) [9], and extracts from elderberry flowers (*Sambucus nigra*) [10] and wormwood (*Artemisia absinthium*) [11].

In the mid-19th century, chemical agents with a broad spectrum of action and high efficacy began to be used, including dichlorodiphenyltrichloroethane (DDT), which was the most widely used [12,13]. Although the use of DDT led to a reduction in insect-borne diseases such as malaria and yellow fever [12], DDT is toxic not only to insects but also to birds and mammals [13,14], so it has been phased out. It has been replaced by organophosphorus compounds and carbamates in crop protection products.

Over the past decade, there has been a change in the amount of insecticide used (Figure 1). In Asia and America, there has been a significant reduction in the use of insecticides. In contrast, Africa, Europe, and Oceania have seen an increase in the use of this group of preparations. However, globally, the amount of insecticide used is decreasing each year, which is in line with the trend of reducing the chemicalization of agriculture and forestry. As a result, the search for natural substances with high selectivity and effectiveness has become a priority in recent years.

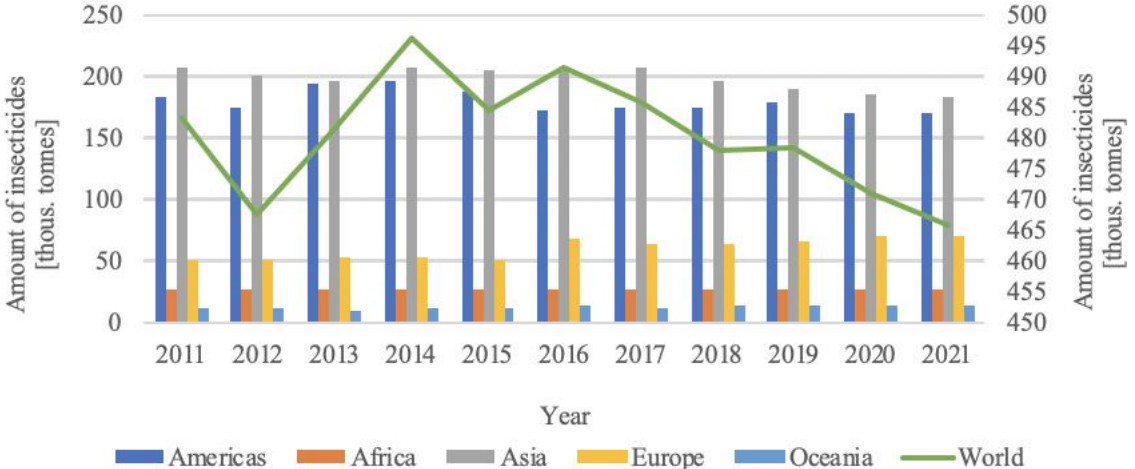

**Figure 1.** Amount of insecticide used in the agricultural sector worldwide (https://www.fao.org/faostat/en/#data/RP, accessed on 15 September 2023).

The purpose of this review is to introduce the reader to the current global issue affecting the forestry sector: the shortage of plant protection products for use against harmful insects. This review outlines the available and used preparations, which are categorized as either synthetic or natural. Additionally, this review presents biological methods of insect control. Oil preparations and the innovative concept of using *Camelina sativa* and *Brassica carinata* plants as substrates in the production of environmentally friendly insecticides are also characterized.

The aim of this review is to demonstrate the potential of natural means of protecting forest stands in the face of the limitation of chemical means of protecting forest ecosystems. The most important section concerns the demonstration of potential new plants as natural ingredients of plant protection preparations with a diversified, modern, and innovative mechanism that repels/kills insects causing the degradation of tree stands.

## 2. Insecticide Characteristics

Insecticides are a vast family of preparations used to control harmful insects. They can be made of natural or synthetic substances [15]. The active ingredients of these preparations can penetrate a pest's body through its epidermis (contact agents) or digestive system (gastric agents). Some agents affect an insect's respiratory system, while others use a

combination of methods. Insecticides that enter a pest's body through the digestive route are administered directly to the plant to enable their preventive effect. There are three types of insecticides, which are segregated based on their modes of action for use in protecting a plant: surface insecticides, depth insecticides, and systemic insecticides. Surface insecticides remain and act on the surface of a plant, depth insecticides penetrate the leaves, and systemic insecticides penetrate the plant and are transported with the sap to all parts of the plant. Insecticides must not exhibit toxicity toward plants. The main classification of insecticides is based on their chemical nature (Figure 2) [16].

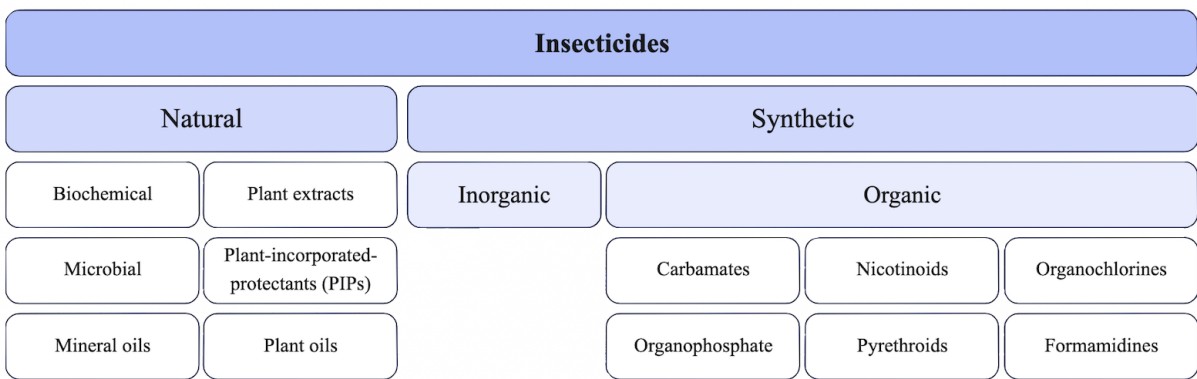

**Figure 2.** Insecticide classification.

## 2.1. Characteristics of Neonicotinoids

Neonicotinoids are a group of highly effective insecticides with a chemical structure similar to nicotine that have been used in crop protection since the 1930s [17]. They are a heterogeneous chemical group, composed of either an open chain (e.g., acetamiprid and clothianidin) or a five- or six-membered ring (thiamethoxam, imidacloprid, and thiachlopride) (Figure 3).

**Figure 3.** Examples of neonicotinoid structures containing a nitroguanidine group and cyanoamidine group.

The active substances can be classified into two main groups: nitroguanidine and cyanoamide (Figure 3). The neonicotinoids belonging to the first group have N-nitro groups in their chemical structures, which contain oxygen atoms and make the molecules much more polar. In the molecules of cyanoamide neonicotinoids, on the other hand, there are cyanoamide groups instead of nitro groups, which do not contain oxygen and are thus much less polar and reactive [18,19]. The chemical structure of the compounds in the two groups mentioned above determines their susceptibility to decomposition in the soil

and metabolism in an insect's body. The biological activity and toxicity of a compound are determined by the type of functional group present in the molecule and its spatial arrangement relative to other substituents.

These compounds can be applied as foliar agents or granules for water and soil or be injected into tree trunks. It is important to note that neonicotinoids should be used objectively and without bias, as their effects on the environment and pollinators are still being studied. Preparations that contain neonicotinoids have a wide range of uses. These compounds have physicochemical properties that allow them to penetrate plant tissues and protect a plant from direct damage by herbivorous insects [20,21].

Neonicotinoids are antagonists of nicotinic acetylcholine receptors, which are the main neurotransmitters responsible for the proper function of the insect brain. Upon entering the body, they replace acetylcholine (ACh) at nicotinic and nicotinic-muscarinic receptors, activating them in manner similar to acetylcholine, with no degradation (Figure 4). This constant stimulation of the receptor interferes with the transmission of chemical signals. Convulsions and spasms caused by neuronal hyperexcitability can lead to the death of an insect [22–24].

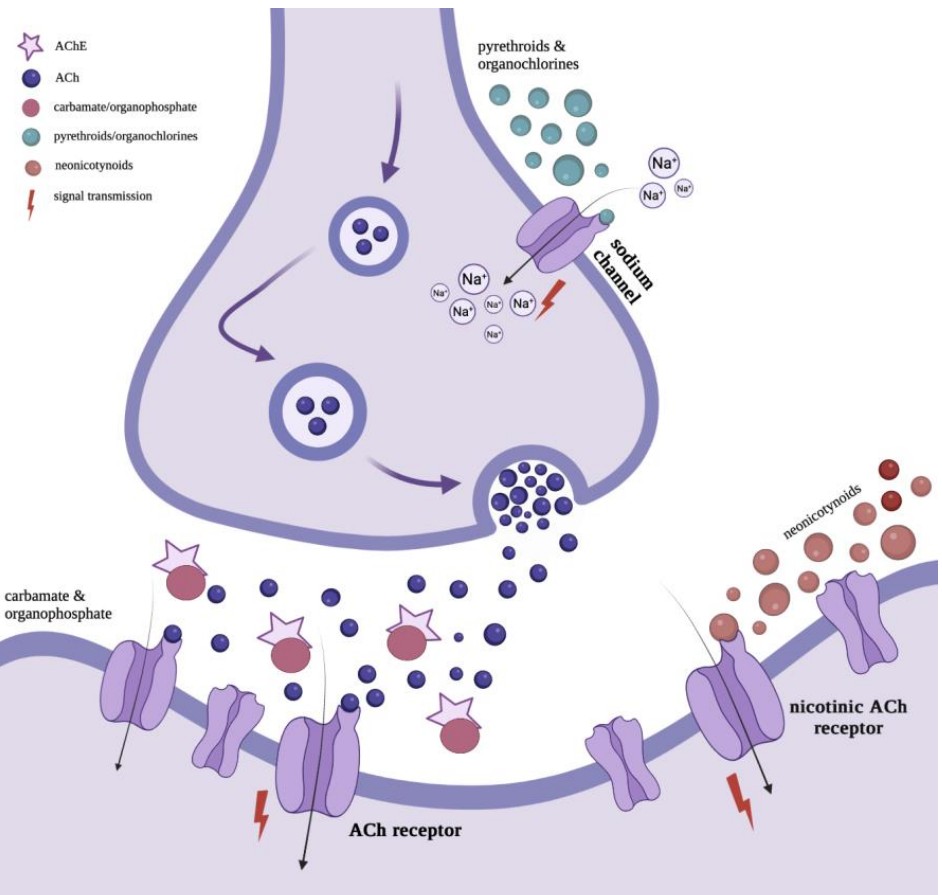

**Figure 4.** Mechanisms of action of different classes of insecticides.

Neonicotinoids are widely used insecticides [25], but some have been restricted due to their harmful effects on the environment. N-nitro neonicotinoids such as imidacloprid and thiamethoxam are particularly toxic [26], posing a high risk to both wild bees as well as honey bees. Even at low concentrations, pesticides can cause disorientation, reduced immunity, and changes in bee reproduction, for which they have been completely banned in the EU [27,28].

### 2.2. Characteristics of Organochlorine Insecticides

Organochlorine insecticides are a group of plant protection products that consist of aliphatic or aromatic hydrocarbons substituted with chlorine atoms. They are divided into three basic groups: dichlorodiphenyl ethanes, cyclodienes, and chlorinated benzenes and cyclohexanes (Figure 5) [29].

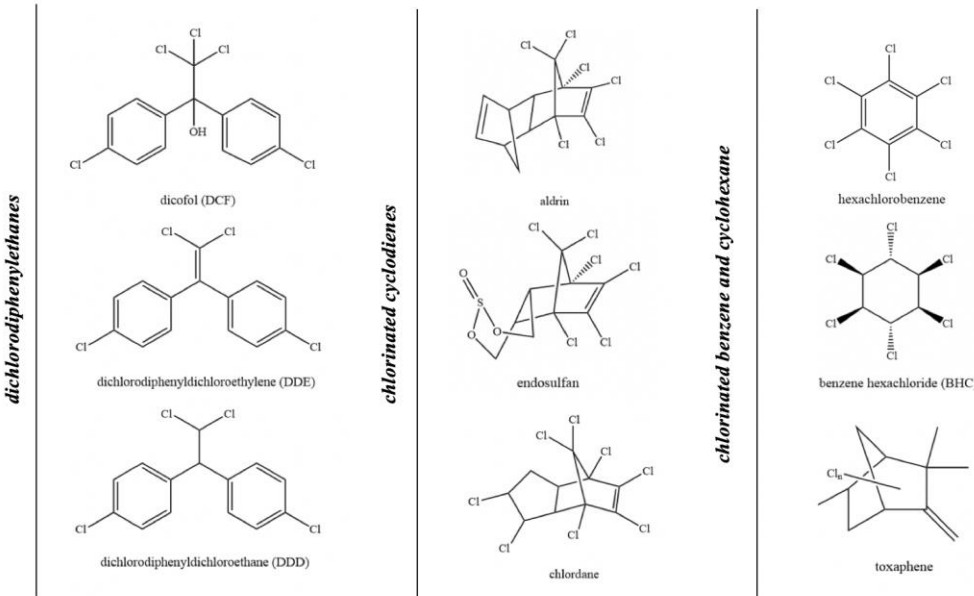

**Figure 5.** Classification of organochlorine insecticides.

These insecticides are known for their high molecular weight, excellent stability, and resistance to photolytic, biological, and chemical degradation, as well as moderate polarity. These compounds exhibit similar physicochemical characteristics, such as persistence, bioaccumulation, and toxicity, due to their structural similarity [30].

The persistence and bioaccumulation capacity of organochlorine compounds can affect the state of the surrounding environment [31]. There are various routes by which they can be released into ecosystems, including as a result of industrial waste disposal and their release from manufacturing plants, absorption by vegetation, and penetration into the soil, consequently polluting groundwater [32]. Given their lipophilic abilities that allow them to bind to human and animal fatty tissues and bioaccumulate in certain organs, such as the liver or kidneys, they are of extraordinary concern [33]. The residence time of organochlorine insecticides in the soil is the best illustration of this potential danger. DDT was found in the soil 8 to 12 years after application, while aldrin and heptachlor were found after 4–12 years [34].

Organochlorine compounds have been banned from production and use in many countries due to their serious consequences, as outlined in the international treaty of the Stockholm Convention. However, in under-industrialized areas such as Africa, their use persists, and they can be transferred to other regions [35,36].

Organochlorine insecticides are a group of neurotoxic poisons that can easily penetrate through an insect's epidermis due to their lipophilic nature. They exert both central and peripheral neurotoxic effects by interfering with the movement of sodium ions in the sodium channels of axons. This causes the ions to leak through the nerve membrane and create a destabilizing negative 'secondary potential', resulting in repeated discharges in the neuron—the so-called 'knock-down' effect (Figure 4). Insects die as a result of the production of endogenous neurotoxin and organism exhaustion [37]. Insecticides that contain chlorinated cyclodienes operate differently. They affect the chloride channels of γ-aminobutyric acid (GABA) receptors, which causes them to open and increases chloride conductance, leading to the inhibition of a new action potential [38].

## 2.3. Characteristics of Organophosphate Insecticides

Organophosphate insecticides such as malathion, parathion, diazinon, and chlorpyrifos are derivatives of phosphoric, phosphonic, phosphosulfuric, or phosphonosulfuric acid, making them attractive alternatives to organochlorine insecticides [39]. These insecticides have a chemically reactive phosphate ester side chain that consists of a central phosphorus atom doubly bonded to an oxygen or sulfur atom and singly bonded to two methoxy (-OCH$_3$) or ethoxy groups (-OCH$_2$CH$_3$) (Figure 6) [40]. These molecules' various positions and numbers of oxygen and sulfur atoms have given rise to several major chemical classes of organophosphorus insecticides, including phosphonates, phosphonates, phosphonites, thionophosphates, thiolophosphates, dithiophosphates, amidophosphates, and pyrophosphates.

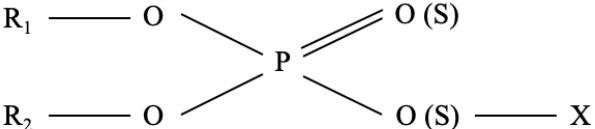

R$_1$, R$_2$ – alkyl substituents
X – acid groups, halogens, phenyl residues, aliphatic and aryl chains

**Figure 6.** Scheme of the structure of organophosphate insecticides.

Organophosphate insecticides are effective due to the irreversible inactivation of the enzyme acetylcholinesterase (AChE), which is essential for normal nerve function in humans, fish, and insects, among other animals. This enzyme is the main cholinesterase that breaks down acetylcholine, which functions as a neurotransmitter, into an inactive form—acetate and choline [41–43]. Thus, the action of organophosphorus compounds is based on inhibition through the phosphorylation of serine in the esterase center of the enzyme, the activity of AChE, and other non-specific esterases (Figure 4). As a result, there is an increase in the concentration of endogenous ACh in the body and its binding to muscarinic and nicotinic receptors in the peripheral as well as central nervous system [44–46]. The consequence of this action is the over-stimulation of the effector organ, which perceives this neurotransmitter as an activating ligand, resulting in the death of an individual [47].

## 2.4. Characteristics of Carbamates

Carbamates are N-methyl esters derived from carbamic acid (NH$_2$COOH) (Figure 7). The first compounds with significant pest control capacity were synthesized in 1954. N-alkylcarbamic acid esters exhibit the strongest insecticidal properties.

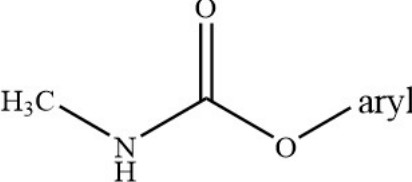

**Figure 7.** Structure of carbamate insecticides.

Carbamate insecticides are unstable and hydrolyze easily, particularly in alkaline environments. Additionally, they exhibit a low level of toxicity, due to which they are often used to destroy insect pests. They act both directly and generally after entering an organism. Their insecticidal activity increases with rising temperatures [34].

The mechanism of action of carbamates is similar to that of organophosphate insecticides. These compounds inhibit the activity of cholinesterases, including AChE, via the carbamylation of amino acid residues in the active site of serine at neuronal synapses [48,49].

This reaction is also part of the metabolic breakdown of carbamates. The insecticide molecule binds to acetylcholinesterase, and, after the formation of a reversible complex, it hydrolytically disintegrates (Figure 4). As a result of this process, the acidic carbamate grouping blocks the catalytic center of the enzyme, and the alcoholic residue is released and undergoes further transformations. Scientists assume that the carbamylation of the enzyme is preceded by several transitional stages. In addition to the esterase center of the enzyme, the anionic center and the alcohol grouping of the insecticide are involved. Due to the low persistence of the reaction products, the carbamylated enzyme is rapidly reactivated, allowing carbamate insecticides to number among the reversible cholinesterase inhibitors [34]. As a result of ACh accumulation, the body's cells are in a state of permanent excitation, resulting in death [48,49].

### 2.5. Characteristics of Pyrethroids

Pyrethroids are characterized by a selectivity of action that is unparalleled in other insecticide groups as well as high pest-killing activity with significantly low toxicity toward humans and other higher organisms. These features have gained them a lot of interest and made them among the most widely used insecticides in the world. Their structure resembles naturally occurring pyrethrins extracted from chrysanthemum flowers (*Chrysanthemum cinerariaefolium*) [50]. Synthetic pyrethroids are esters of chrysanthemic acid and alcohols of various structures. Usually, they are a racemic mixture of stereoisomers. Due to their chemical structure and mode of action, we can divide them into two groups (Figure 8): type I pyrethroids, which have cyclopropanecarboxyl in their structure and cause hyperactivity and deterioration of an insect's coordination, as well as type II pyrethroids, containing an α-cyano group. The latter group potentiates their insecticidal effect because it induces depolarization of the nerve, thus causing the paralysis of an individual [51–53].

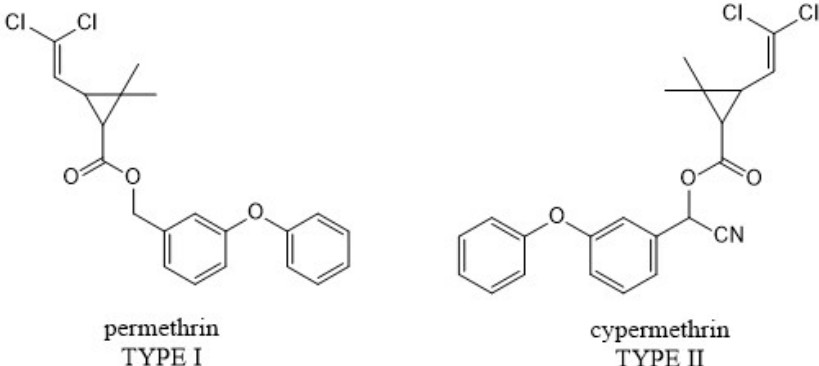

**Figure 8.** Chemical structure of type I and type II pyrethroids.

Pyrethroid insecticides can enter organisms through the digestive tract, respiratory tract, and epithelium. Their lipophilic properties facilitate their spread in an insect's body and transport to the nervous tissues, where they interfere with the excitation process and conduction of action potential, thus disrupting the proper functioning of the peripheral nervous system. The operation of pyrethroid compounds mainly involves altering the dynamics of ion flow in sodium channels to prevent their closure, resulting in a constant and unrestricted flow of sodium ions. This leads to permanent depolarization of the axial membrane and paralysis of an insect's body (Figure 4) [54,55]. Additionally, high concentrations of pyrethroid compounds can disrupt the function of GABA-gated chloride channels, leading to increased seizures [56].

Pyrethroids are commonly used as insecticides in forestry, horticulture, and agriculture. Although they are considered relatively safe, their widespread use can lead to accumulation in the environment, which may be harmful to humans and animals [57].

### 3. Natural Protective Products for Protecting Plants against Insects

Plant protection against insects relies on four main pillars: prevention and biological, chemical, and physical action. The use of chemical insecticides, a common practice, should be avoided unless all other methods have failed due to their potential negative impact on both humans and the environment.

#### 3.1. Mechanical Methods

Pests can be removed from plants through physical means, such as spraying a plant with water or manually picking insects off. Traps and barriers can also be employed to protect a plant from an insect attack. Placing sticky bands on tree trunks is a commonly used physical method [58].

#### 3.2. Biological Methods

Biological control can be used alone or in combination with other control methods in IPM (Integrated Pest Management) programs. It focuses on using natural enemies to reduce populations of harmful insects without the use of synthetic insecticides [59,60]. There are three main approaches to biological control:

1.  Augmentative biological control—increasing the density of native or non-native natural enemies through regular releases;
2.  Conservation biological control—the manipulation of a habitat to increase the reproduction, survival, and effectiveness of natural enemies already present in the affected area;
3.  Classical biological control (CBC)—the introduction of a natural enemy of native origin to control a pest, which is usually also non-native, to determine whether the population of the natural enemy is sufficient to achieve permanent control of the target pest.

#### 3.3. Bioinsecticides

Bioinsecticides are a competitive category of insecticides that include naturally occurring compounds or agents derived from microorganisms, plants, and animals. They inhibit the growth and rapid spread of insect pests through various mechanisms of action (in addition to disrupting the nervous system) [61–63]. These preparations do not pose a threat to living beings and do not adversely affect the environment. Sustainable agroforestry management based on biopesticides is socially accepted, promotes economic productivity, and contributes to environmental stewardship, constituting the basis of sustainable development [64].

Bioinsecticides can be classified based on their origin and the type of compound required to form an effective formulation. The classification includes microbial, biochemical (such as essential oils, plant extracts, insect growth regulators, and insect pheromones), and plant-incorporated protectants (PIPs) [65,66]. Currently, microbial biopesticides are the most significant. Biopesticides are widely used due to their low toxicity, selectivity of action, and ability to be easily biodegraded [67]. Transgenic plants (PIPs) offer an attractive alternative. They contain molecules such as Bt Cry proteins, $\alpha$-amylase inhibitors, or double-stranded ribonucleic acid (dsRNA), which deter insect pests from targeting these plants [68–70].

Over the past decade, the proportion of biopesticides used in crop protection products has varied greatly (Figure 9). In 2020, there was a significant increase in the use of botanical insecticides, although they remain a niche category of crop protection products. This is primarily due to their high production cost and the labor- and time-consuming process of introducing them to the market. Additionally, their limited durability and effectiveness are correlated with changing atmospheric conditions. Furthermore, they have targeted action against a specific pathogen species [62].

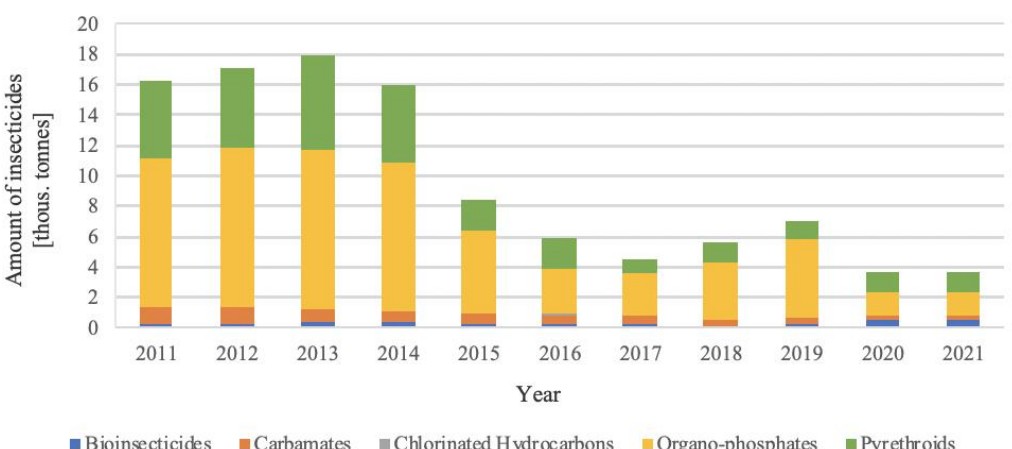

**Figure 9.** Share of particular classes of insecticides in Europe. (https://www.fao.org/faostat/en/#data/RP, accessed on 20 September 2023).

### 3.4. Oils as Botanical Pesticides

An important issue in the chemical control of insects is the presence of pesticide residues in the environment. This depends not only on the type of pesticide used but also on the dosage, the number of treatments, the form of preparation, the type of plant and soil, and weather conditions. As the product will be subject to runoff or be washed away by rain after treatment, penetrate deep into a plant, and accumulate in the mulch, it is extremely important to observe the withdrawal period indicated on the label. In this respect, botanical insecticides are a promising alternative to conventional products. They usually have a relatively short withdrawal period, and their use does not put non-target organisms at risk. They do not tend to accumulate in the soil because they degrade quickly. Their use also reduces the likelihood of insects developing resistance. These characteristics are making botanical insecticides increasingly popular. This relationship is illustrated by the frequency of searches for combinations of phrases such as "botanical extract" and "insecticide", which almost tripled between 2012 and 2022, increasing from 1158 to 3377 reports, and this number is steadily increasing (Scopus database, https://www.scopus.com, accessed on 29 November 2022).

Among biopesticides, plant extracts are important components in addition to insect pheromones, microbial pesticides, or insect growth regulators. These are naturally occurring insecticides in plants containing a number of bioactive phytochemicals [71]. They are found in the form of plant oils and essential oils. The former are obtained by extracting or pressing seeds and other plant parts, while essential oils are produced through distillation [72].

### 3.4.1. Essential Oils

Essential oils are fragrant mixtures of volatile organic compounds synthesized by plants as secondary metabolites. They are found in plants or their individual parts—such as flowers, leaves, fruits, roots, or seeds. They are localized in specific structures of secretory tissues (glandular hairs, secretory cavities, and resin ducts), where they accumulate as final products of metabolism [73,74]. Chemically, oils are complex mixtures of different compounds, the composition of which is not always completely known and is often variable. Usually, the dominant compound imparts a fragrance to the oil. The main constituents of essential oils are terpene compounds: mono-, sesquioxin-, and diterpenes (terpene oils) and phenylpropane derivatives (non-terpene oils). The compounds they contain are hydrocarbons, alcohols, aldehydes, ketones, esters, and ethers [75]. The composition of the oil also depends on the part of the plant from which it is obtained. An example is bitter orange (*Citrus aurantium*), which produces three different oils in young shoots, leaves, and flowers [74,76,77].

The compounds contained in essential oils have been used as flavorings in the food and perfume industries but also in medicine, mainly as medicinal herbs, and as an element of aromatherapy. There is growing interest in their use as natural plant protection agents [78]. The modes of action of essential oils in this area are very diverse. They can interfere with the gas exchange process in insects, as well as their ability to identify host plants, and egg laying [79–81]. Some also act as antifidants, attractants, or repellents. To a large extent, they affect the hormonal balance, causing malformations, overstimulation, and consequently death in insects [82,83]. Essential oils can also form mixtures with synthetic insecticides, increasing their effectiveness [84]. Examples of essential oils extracted from different insecticidal plants are shown in Table 1.

**Table 1.** Essential oils of some plant families with insecticidal efficacy against different insect pests.

| Plant | | | Insect | | |
|---|---|---|---|---|---|
| **Family** | **Name** | **Major Constituent(s)** | **Order** | **Name** | **Ref.** |
| Apiaceae | *Foeniculum vulgare* | estragole | Coleoptera | *Tribolium castaneum* | [85] |
| | *Coriandrum sativum* | linalol, geranyl acetate | | *Tribolium confusum* | [86] |
| | | | | *Callosobruchus maculatus* | |
| Asteraceae | *Artemisia capillaris* | 1,8-cineole, germacrene D, camphor | | *Sitophilus zemais* | [87] |
| | *Artemisia mongolica* | α-pinene, germacrene D, γ-terpinene | | | |
| | *Echinacea purpurea* | β-cubebene, caryophyllene | | *Sitophilus granarius* | [85] |
| Ericaceae | *Ledum palustre* | ascaridole, p-cymene | Diptera | *Culex quinquefasciatus* | [88] |
| | | | | *Musca domestica* | |
| | | | Lepidoptera | *Spodoptera littoralis* | |
| Lamiaceae | *Romarinus officianlis* | 1,8-cineole, geraniol | Coleoptera | *Sitophilus oryzae* | [89] |
| | *Ocimum basilicum* | methyl cinnamate, linalool, eucalyptol | | *Sitophilus granarius* | [85] |
| | | | Lepidoptera | *Spodoptera litura* | [90] |
| Lamiales | *Mentha pulegium* | pulegone, β-pinene, linalol, eucalyptol | Coleoptera | *Sitophylus oryzae* | [89] |

Essential oils are considered a very promising group of compounds due to their high efficacy and low toxicity toward vertebrates. Their low persistence under field conditions increases the safety of farmers' work and reduces the likelihood of residual compounds being present in food products. The disadvantages of using essential oils are their relatively low activity, requiring the application of high concentrations, and an intense odor that affects the quality and desirability of the plant product [78,82].

### 3.4.2. Plant Oils

Contact formulations based on vegetables oils, which are characterized by limited persistence, are an alternative to the use of conventional insecticides. Their precursors were paraffin-oil-based formulations derived from petroleum refining, which showed high insecticidal efficacy. The main limitation to their use was their high phytotoxicity. This was largely reduced by selecting distillates with specific boiling ranges and appropriate refining parameters. The multi-stage and labor-intensive production process, increased raw material costs, and potential toxic effects on plants prompted the search for more environmentally friendly substrates. Following this trend, the combination of highly refined paraffinic oil with vegetable oils seemed to be the solution. This resulted in formulations characterized by high efficacy and low phytotoxicity.

Their main mechanism of action is based on the physical blocking of insects' respiratory tracts (fistulas). The oil preparation applied to them forms a thin, hydrophobic layer and prevents gas exchange, resulting in suffocation. They also interfere with insects' ability to feed on oil-coated surfaces. In some cases, oils can also interact with insects' fatty acids, disrupting their metabolism [91]. Oils can be mixed with other insecticides to provide a broader spectrum of activity and longer-lasting formulations. The insects most commonly controlled in this way include spider mites, whiteflies, and juvenile scales. Vegetable oils also perform well as preparations for preventing the transmission of viruses by insects and as fungicides (e.g., against powdery mildew) [92].

The most common ingredient in oil-based insecticide formulations is oil from the seeds and bark of the Indian honeybush (*Azadirachta indica*). This oil is readily available, relatively inexpensive, and, most importantly, does not harm birds or other beneficial insects. It owes its action to triterpenes, compounds from the limonoid group, the main representative of which is azadirachtin. It is described as an antifidant compound that interferes with the feeding and reproductive abilities of insects [93]. An affected insect does not die directly as a result of azadirachtin entering the body but as a result of the loss of the ability to forage. The other compounds in neem oil interfere with an insect's normal development (molting and growth) and act as a deterrent against them [94,95]. The oil of *Pongamia pinnata* (L.) *Pierre* is also becoming increasingly important, containing about 5% flavonoids, of which 2% is the furanoflavonoid caranjin, which is responsible for the insecticidal activity of the oil [78]. Other common ingredients used in commercial insecticidal formulations are cottonseed oil and soybean oil.

The described ingredients included in natural preparations intended to repel insects feeding on tree stands are difficult to isolate, which makes them expensive and inefficient. It would be unprofitable to use them in forest stands. Therefore, it is necessary to look for cheaper and more effective preparations of plant origin, hence the idea to use oils from *Camelina sativa* and *Brassica carinata*. These oils are rich in bioactive compounds and have a hitherto unexplored potential to repel/kill insects that cause the degradation of tree stands.

## 4. New Potential Botanical Insecticides

Oils pose little risk to humans and most animals, including beneficial enemies of insect pests. Plant protection products based on them are therefore an attractive alternative to chemical formulations, and the possibility of integrating them with biological control mechanisms makes it possible to achieve highly satisfactory efficacy. Following this trend, it is worth looking at lesser known and under-utilized oilseed crops such as *Brassica carinata* and *Camelina sativa*. Their growth requirements are not too prohibitive, making them suitable for growth on infertile land. This is an important consideration, as access to productive arable land is becoming increasingly limited. The plants are mainly used as a feedstock for biofuel production, but the high oil content of their seeds means their potential could be much wider.

*Camelina sativa* is native to Europe and Central Asia. It is known by various names, such as false flax, gold-of-pleasure flax, and Dutch flax [96]. It is considered a weed in most parts of the world, although in Poland, the castor oil extracted from its seeds is a valuable part of individuals' diets. Health Canada has also approved its use for food purposes [97]. Unlike camelina, carinata, also known as Ethiopian mustard, is a non-food crop. The oil extracted from its seeds is characterized by high levels of erucic acid, which is harmful to humans, and its consumption can, for example, contribute to myocardial lipidosis [98–100]. The residue from oil pressing, on the other hand, is used as a high-protein, low-fiber feed additive for livestock [101,102]. Both plants are characterized by their ability to grow in harsh semi-arid conditions with limited water availability. Canada and the United States are keen to use them as cover crops to prevent soil erosion, nutrient leaching from the soil, and weed overgrowth [103–105].

### 4.1. Characteristics of Physicochemical Properties of Brassica carinata and Camelina sativa

The potential of the aforementioned plants is primarily attributed to the high oil content of their seeds. Depending on the source, carinata seeds contain between 33% and 51% oil, while camelina seeds contain between 30% and 49%. Additionally, these seeds are rich in protein, with carinata and camelina seeds containing 20%–38% and 24%–32% protein, respectively, as well as other bioactive compounds [106–108]. Differences in the content of individual components are primarily caused by variations in varieties, plant maturity, and the prevailing climatic conditions during growth [109].

*Camelina sativa* oil, commonly referred to as camelina oil, is a unique vegetable fat. It has a distinct taste and smell, often compared to onions or mustard. The oil has a

dark-golden color, sometimes appearing orange. In contrast, carinata oil is yellow and has a noticeable aroma [110].

The refractive index (RI) is a crucial parameter in oil characterization as it correlates with the relationship between fatty acid chain length, molecular weight, and the degree of unsaturation. Low RI values indicate the presence of long fatty acid chains, while high values indicate a high number of double bonds in the fats studied [110]. In the cited case, both oils analyzed had an RI of 1.47 (Table 2), tentatively confirming that they are highly unsaturated substances. The oils' highly unsaturated fatty acids may undergo oxidative degradation processes due to the configuration of double bonds [111]. Therefore, it is likely that secondary products of oil oxidation were present in the tested material, which may distort the obtained result. As a result, verification using other techniques is necessary [112].

**Table 2.** Physicochemical and quality parameters of *Camelina sativa* and *Brassica carinata* oils [105,107,110].

| Parameter | *Camelina sativa* | *Brassica carinata* |
|---|---|---|
| Specific gravity (25 °C) | 0.91–0.92 | 0.90–0.95 |
| Refractive index (20 °C) | 1.47 | 1.47 |
| Iodine value (g $I_2$/100 g oil) | 143.18–162.26 | 90.00–113.00 |
| Saponification value (mg KOH/g oil) | 178.60–187.80 | 129.00–154.0 |
| Unsaponifiable fraction (%) | 0.54–0.87 | 4.20–6.60 |
| Peroxide value (meq $O_2$/kg oil) | 0.89–3.47 | 4.10–9.10 |
| *p*-Anisidine value | 0.22–1.48 | 3.10–6.30 |
| TOTOX number | 2.16–8.10 | 11.40–24.50 |

The iodine value (IV) is a commonly used parameter for determining the degree of unsaturation of fat/oil. It represents the mass of iodine atoms attached to the C=C double bonds of fatty acids. The higher the level of unsaturation, the more iodine atoms are attached to the molecule, and the higher the iodine number [113,114]. Additionally, it is used to evaluate the effects of oxidation reactions involving mono- and polyenoic fatty acids. Coconut oil is a fat that contains a high percentage of saturated fatty acids, resulting in an iodine number range of 8–10 g $I_2$/100 g of oil [115]. Table 2 summarizes the iodine numbers of oils extracted from *Camelina sativa* and *Brassica carinata* seeds. The values shown indicate the unsaturated nature of both oils, with camelina oil's IV standing out significantly (143.18–162.26 g $I_2$/100 g of oil), which is far superior to sunflower oil's IV (118–141 g $I_2$/100 g of oil) [116]. Furthermore, it can be observed that camelina and carinata oils are mainly composed of long-chain fatty acids, as demonstrated by their low saponification value (SV) (Table 2). This value is similar to the same values for canola oil (163 mg KOH/g of oil) and sunflower oil (183 mg KOH/g of oil) [117,118].

Carinata oil contains approximately seven times more unsaponifiable fractions than oil from gold-of-pleasure flax (Table 2). The unsaponifiable fraction comprises sterols, aliphatic alcohols, terpene alcohols, hydrocarbons, phenols, and pigments [119]. It is characterized by its anti-inflammatory and antioxidant effects, which have a positive impact on oil stability [120,121].

The stability of fat is heavily influenced by its oxidation processes. The peroxide number (PV) is an indicator used to determine the quantity of primary oxidation products in oils, as shown in Table 2. *Brassica carinata* has a higher percentage of primary oxidation products compared to *Camelina sativa*. It is important to consider other factors as well. However, it is not possible to draw clear conclusions about the oxidation state of an oil based solely on the peroxide content. A useful indicator is the anisidine number, which determines the content of aldehydes and ketones in the oil under study. These are the products of the decomposition of peroxides and hydroperoxides. The TOTOX index is the culmination of oil stability studies and allows for the determination of the overall degree of oxidation of an oil [122]. According to the data presented in Table 2, it can be concluded that the oil from gold-of-pleasure flax is more stable than the oil from carinata. This is supported by the significantly lower TOTOX index value.

*4.2. Fatty Acid Compositions of Brassica carinata and Camelina sativa*

Oils from *Camelina sativa* and *Brassica carinata* are rich sources of unsaturated fatty acids, with differing compositions. Table 3 shows that gold-of-pleasure flax oil is mainly composed of polyunsaturated fatty acids, while monounsaturated acids predominate in *Brassica carinata*. Both analyzed fats contain a small percentage of saturated fatty acids, which make up around 10% of total fatty acids.

Researchers have conducted a series of analyses and identified 18 fatty acids in camelina oil. The four fatty acids with the highest percentages are eicosenoic acid, linoleic acid, α-linolenic acid, and oleic acid. α-linolenic acid is the dominant fatty acid, with a content range of 28.0%–50.3%. Camelina oil, along with flaxseed oil, is a potential source of plant-based omega-3 fatty acids in diets, as the proportion of this acid in typical oilseeds such as sunflower and olive oil is negligible and rarely exceeds 1% [113,123]. Additionally, this oil contains eicosenoic acid and eruic acid, which are not commonly found in vegetable oils but are typical of oils extracted from plants of the Cruciferae family [124]. Camelina oil is a source of medium-chain fatty acids, which were previously only obtained from palm and coconut oils [125]. These fatty acids have been found to inhibit oxidation and fat deposition in both humans and animals [126,127]. This fact makes camelina oil a valuable addition to any diet.

**Table 3.** Saturated and unsaturated fatty acid compositions of *Camelina sativa* and *Brassica carinata* oils [105,107,124].

|  | *Camelina sativa* | *Brassica carinata* |
|---|---|---|
| PUFA [%] | 50.10–72.00 | 17.30–36.90 |
| MUFA [%] | 17.40–41.40 | 52.80–71.00 |
| SFA [%] | 9.10–13.12 | 4.80–11.00 |
| ω-6/ω-3 | 1:1.51–2.87 | 1: 0.80–2.80 |

PUFA—polyunsaturated fatty acids, MUFA—monounsaturated fatty acids, and SFA—saturated fatty acids.

Seventeen fatty acids have been quantified in oils obtained from *Brassica carinata*, with erucic acid being the most abundant, followed closely by oleic acid, linoleic acid, and linolenic acid [110]. However, eruic acid is also a valuable raw material in the production of amphiphilic compounds, plasticizers, and polymers. Therefore, genotypes with the highest possible erucic acid content are also being developed [128,129]. *Brassica carinata* oil was found to contain elaidic acid, pailinic acid, eicosapentaenoic acid, and docosavtetraenoic acid, which are not present in camelina oil [130]. Table 4 shows the complete fatty acid profiles of the oils analyzed. The content of individual fatty acids in these oils varies depending on the variety, location, environment, and extraction method used.

**Table 4.** The fatty acids composition (%) of *Camelina sativa* and *Brassica carinata* oils [107,110,124–127].

| Fatty Acids [%] | *Camelina sativa* | *Brassica carinata* |
|---|---|---|
| C12:0 Lauric | 0.04–0.05 | – |
| C14:0 Myristic | 0.13–0.16 | 0.02–0.08 |
| C16:0 Palmitic | 5.10–6.59 | 2.30–4.10 |
| C16:1 Palmitoleic | 0.10–0.14 | 0.04–0.95 |
| C17:0 Margaric | 0.06–0.10 | 0.01–3.20 |
| C18:0 Stearic | 2.19–3.42 | 0.73–3.17 |
| C18:1n9 cis Oleic | 14.90–20.47 | 7.00–29.50 |
| C18:1n9 trans Elaidic | – | 0.03–7.60 |
| C18:2n6 cis Linoleic | 16.00–22.40 | 9.10–21.79 |
| C18:3n3 Linolenic | 28.00–50.30 | 4.70–19.30 |
| C20:0 Arachidic | 1.37–1.80 | 0.50–1.20 |
| C20:1n11 cis Paillinic | – | 1.30–11.00 |
| C20:1n9 Eicosenoic | 11.51–17.50 | 0.55–5.10 |

**Table 4.** *Cont.*

| Fatty Acids [%] | *Camelina sativa* | *Brassica carinata* |
| --- | --- | --- |
| C20:2n6 Eicosadienoic | 0.30–2.00 | 1.00–1.07 |
| C20:3n3 Eicosatrienoic | 1.14–3.10 | – |
| C20:5n3 Eicosapentaenoic | – | 0.20–1.70 |
| C22:0 Behenic acid | 0.27–0.80 | 0.25–1.41 |
| C22:1n9 Eruic | 1.52–4.23 | 20.1–56.3 |
| C22:2n6 Docosadienoic | 0.09–0.17 | – |
| C22:4n6 Docosatetraenoic | – | 0.68–2.0 |
| C24:0 Lignoceric | 0.13–0.20 | – |
| C24:1n9 Nervonic | 0.42–0.90 | – |

*4.3. Composition of Unsaponifiable Fraction of Brassica carinata and Camelina sativa*

The high stability of oils is greatly influenced by the presence of natural antioxidants in addition to the fatty acid composition. Tocopherols are the secondary lipid components that are present in oils. Their total content in camelina oil is 410–800 mg/kg depending on the source [108,123,131]. An analysis by Günç Ergönül [132] showed the presence of α-tocopherol, β-tocopherol, γ-tocopherol, and δ-tocopherol in camellia oil. In each source, the content of γ-tocopherol was the highest. Its high concentration ensures a reasonable shelf life for the oil. No special storage conditions are required. Information on the tocopherol content of Brassica carinata oils is limited. According to Velasco [133], the total value of tocopherols is between 96 and 196 mg/kg, and the ratio of α/γ-tocopherol is on average 1.0.

Sterols are present in all eukaryotic organisms and serve as the fundamental building blocks of cell membranes. They regulate various metabolic processes, such as signal transmission and the activity of membrane-bound enzymes. Additionally, they serve as precursors to steroid hormones in humans and as brassinosteroids, which are involved in important plant development processes. Kurasiak-Popowska [127] identified six sterols, including brassicasterol, β-sitosterol, campesterol, cholesterol, sitosanol, and δ-5,24-stigmastadienol, with β-sitosterol being the most abundant. This confirms the results of the analyses conducted by Shukla [134]. The total sterol content in camelina oil is estimated to be between 3600 and 5900 mg/L [123].

The chlorophyll pigment composition and content depend on seed maturity and atmospheric conditions during plant cultivation. The level of chlorophyll pigments in camelina oil varies between 0.60 and 2.60 mg/kg [135]. Compounds from the carotenoid group may facilitate oil oxidation in the presence of light. However, most available studies only report the total carotenoid content of camelina oil, which ranges between 8.11 and 112 mg/L [136,137]. The analysis primarily focuses on determining the levels of β-carotene. However, Kurasiak-Popowska [127] estimated not only β-carotene levels (129.56 mg/L) but also the levels of lutein (12.16 mg/L) and zeaxanthin (1.78 mg/L).

Vegetable oils are a valuable source of bioactive compounds. Research on them has mainly focused on the analysis of lipophilic compounds such as sterols, tocopherols, and carotenoids. However, there is a growing trend towards studying the previously underestimated profile of phenolic compounds. These compounds play a significant role in maintaining the stability of an oil by preventing lipid oxidation in radical reactions and contribute to the sensory properties of a substance. Günç Ergönül [132] identified several phenolic compounds in camelina oil: tyrosol, apigenin, syringic acid, 3-hydroxytyrosol, luteolin, p-coumaric acid, ferulic acid, 4-hydroxybenzoic acid, sinapic acid, vanillic acid, veratric acid, and caffeic acid. Kurasiak–Popowska [127] also observed the presence of four additional phenolic compounds: chlorogenic acid, gallic acid, protocatechuic acid, and t-cinnamic acid. It was observed that the oil contained more phenolic acids than the studied pomace. However, Terpinc [109] drew a different conclusion, stating that most of the phenolic compounds did not transfer into the oil. The only compounds present in the oil were small amounts of catechins, p-hydroxybenzoic acid, ellagic acid, salicylic

acid, and quercetin. The content of phenolic compounds in camellia oil ranges from 990 to 1536 mg/100 g.

*4.4. Insecticidal Properties of Compounds Present in the Tested Oils*

Plants have adapted to function in adverse environmental conditions by synthesizing phytochemicals such as phenols, terpenoids, and alkaloids. These compounds can act collectively or independently to protect plants against herbivores, pathogens, and insect pests. Considering the issue at hand, there has been a focus on the potential of isolating individual compounds or utilizing plant extracts as potent agents to combat insecticide-resistant insect pests.

Several publications have demonstrated the efficacy of plant metabolites in negatively impacting key physiological processes of insects. In a recent study, a team of South Korean scientists investigated the insecticidal activity of compounds extracted from the bark of the fragrant cinnamon tree (*Cinnamomum cassia*) against the fourth-instar larvae of *Mechoris ursulus*. The filter paper diffusion method was used as a substrate to release the analyzed compounds. The study found that t-cinnamic acid at a dose of 5 mg/filter paper had a strong insecticidal effect (86.7%). Additionally, it was demonstrated that at the same concentration, t-cinnamic acid exhibited high fumigation activity against larvae, resulting in 77.8% mortality [138]. Murillo [139] extracted compounds from the aboveground parts of *Piper septuplinervium* and inflorescences of *Piper subtomentosum* Trel. & Junck. A study of insecticidal activity against *Spodoptera frugiperda* (Lepidoptera: Noctuidae) showed high insecticidal activity of protocatechuic acid, whose LC50 was determined to be 17.16 ppm. Protocatechuic acid was also toxic to *Spodoptera litura* (Lepidoptera: Noctuidae) larvae. A median lethal dose of LC50~6.01 mg/mL was determined. In addition, its insecticidal activity was found to be due to the inhibition of carboxylesterase activity [140]. Gallic acid is the primary phenolic acid found in both the flesh and peel of mangoes. It serves as a constitutive protection against *Bactrocera dorsalis* (Hendel). In laboratory studies, Shivashankar demonstrated that gallic acid acts as a pro-oxidant in insect bodies [141]. Due to its strong toxicity against *Bactrocera dorsalis* (Hendel), it can be used as part of an IPM strategy. Numerous reports exist on the insecticidal activity of chlorogenic acid. A study by Wang [142] showed that chlorogenic acid was the most toxic of the chemical insecticides tested against *Bemisia tabaci*. The application of the LC25 dose resulted in longer insect developmental stages, slower pupation, and reduced female fecundity, thereby altering the sex ratio in populations. The study also concluded that chlorogenic acid exhibits sublethal effects, which are evident in the reduced survival rates of nymphs, pseudopupae, and adults. These results support the findings in Lin's study [143], which demonstrated that the application of an LC20 dose of chlorogenic acid prolonged larval growth. The addition of chlorogenic acid (5-O-caffeoylquinic acid) to the medium for *Spodoptera litura* F. larvae resulted in decreased weight and a twofold increase in mortality compared to the control. This study confirms that chlorogenic acid is safe for beneficial organisms and humans. Summers [144] noted that chlorogenic acid and caffeic acid extend the developmental time of *Helicoverpa zea*. Additionally, caffeic acid was found to induce conformational changes in intestinal proteases in *Helicoverpa armigera* [145], leading to the inhibition of their activity and intensifying their insecticidal action. Furthermore, studies have demonstrated that the consumption of caffeic acid by *Spodoptera litura* F. reduces nutritional parameters, leading to a decrease in digestive efficiency and ultimately resulting in a slower growth rate of the insect [146]. Research conducted on the second developmental stage of *Spodoptera litura* larvae has shown that their ingestion of caffeic acid leads to a reduction in body weight. The LC50 value was determined to be 385.19 ppm.

Several studies suggest that flavonoids play a significant role in protecting plants against insect pests. Herrera-Mayorga [147] demonstrated the insecticidal activity of quercetin, the main secondary metabolite found in the leaves of *Solidago graminifolia*, against *Spodoptera frugiperda*. Quercetin exhibited strong insecticidal activity against third- and fifth-stage *Spodoptera frugiperda* larvae, with an LC50 value of 0.157 mg/mL. Quercitin's

mechanism of action in the insect body has been shown to consist of the inhibition of AChE with better binding values than acetylcholine. Similar action is demonstrated by the phytochemically active apigenin, which binds and inhibits AChE1 in *Cx. Quinquefasciatus*, inducing larval death [148]. It has been shown to damage the cylindrical cells of the midgut epithelium, thereby causing significant deformities within these cells. Extracts from *Olea europaea* subsp. *laperrinei*, containing flavonoids and caffeic acid, have been found to disrupt adult fecundity, shorten the laying periods of females, and prolong pupal development. Additionally, quercitin, apigenin, and oleuropein have been identified as being directly responsible for the insecticidal effects on *Ephestia kuehniella* [149]. Furthermore, Goławska [150] observed that apigenin has an antifeedant effect. Sustained high levels of apigenin in *Medicago sativa* leaves result in a significant reduction in *Acyrthosiphon pisum* levels in the plant. This is due to the inhibition of phloem sap uptake by insects, leading to a reduction in pea aphid abundance on the plant.

Plant resources are a vast source of bioactive substances that could potentially address the current environmental sustainability issues related to crop protection products. Research suggests that selecting specific compounds or plant extracts can lead to the development of effective plant-derived insecticides that align with the IPM strategy.

## 5. Conclusions

Among the factors discussed, insect pests are the most prominent. Biotic factors pose a significant threat to forests, which are an important element of the natural environment. According to data released by Forest Europe in 2020, as much as 3% (5053 ha) of Europe's forests have been damaged due to the presence of insects, and this number continues to rise. Until now, the most effective method of combating insect pests was considered to be the use of chemical insecticides, the residues of which pose many long-term threats to both land and water ecosystems, as well as living organisms. The methods described in this paper for combating forest pests, ranging from chemical to biological approaches, allow for the systematization of knowledge on this subject. The advantages and disadvantages of the preparations used so far have been indicated. The key element of this work is the proposal of the use of two plants as potential resources that exhibit enormous biological potential. The rich composition of *Brassica carinata* and *Camelina sativa* seeds predisposes this resource to the production of biopesticides with a broad mechanism of action. On the one hand, the oil will have a physical effect, while the bioactive compounds dissolved or suspended in it, such as extracts of fat-insoluble compounds, may have repellent or insecticidal properties. This paper presents evidence of the potential effects of these substances. A comprehensive compendium of knowledge on insecticides has been prepared and can be used by other researchers who are practically engaged in addressing the described issues. The research team that conducted this literature review will proceed with further research on the development of a preparation based on oils and extracts obtained from *Brassica carinata* and *Camelina sativa* seeds.

**Author Contributions:** Conceptualization, K.R., K.S.-S. and D.K.-P.; software, K.R.; resources, K.R. and K.S.-S.; data curation, K.R.; writing–original draft preparation, K.R. and K.S.-S.; writing–review and editing, K.R., K.S.-S. and D.K.-P.; visualization, K.R. and K.S.-S., supervision, K.S.-S. and D.K.-P.; funding acquisition, K.R., K.S.-S. and D.K.-P. All authors have read and agreed to the published version of the manuscript.

**Funding:** This research was funded by the European Union's Horizon Europe research and innovation program under grant agreement No. 101081839.

**Data Availability Statement:** Data sharing is not applicable to this article.

**Conflicts of Interest:** The authors declare no conflict of interest.

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
