# Peer review of "Diverse Approaches to Insect Control: Utilizing Brassica carinata (A.) Braun and Camelina sativa (L.) Crantz Oil as Modern Bioinsecticides"

_forests, doi:10.3390/f15010105_

Round 1
Reviewer 1 Report
Comments and Suggestions for Authors
The manuscript has 18% plagiarism needs to be fully corrected. File is attached for your reference

Author Response
Thank you very much for your valuable comment. It have been considered and revised meticulously.
Answer: The text has been redrafted in accordance with the Reviewer's suggestions.
Reviewer 2 Report
Comments and Suggestions for Authors
This review introduces global problem currently facing the forestry and agricultural sector, namely - the shortage of plant protection products against harmful insects. It describes the advantage and disadvantage of synthetic and natural insecticides. Moreover, two plants namely Camelina sativa and Brassica carinata plants have been recommended for future environmentally friendly insecticides.
Analyses of this review suggested following revisions:
-Full form for the abbreviation should be provided when they are being used for the first time. E.g EU in abstract
-Botanical name of the Plants should be in Italics.
-So many English and typographical mistakes through out the manuscript should be rectified
- Authors should provide a complete, structured and systematic summarization on the related key aspects.
-Authors should outline detailed views on future research directions and perspectives.
- The numbering of sections and subsections are problematic and must be rectified. A index must be provided for the readers
- Why authors are recommending only two plants for future studies should be clearly explained. What about other plant extracts and essential oils in the literature as bioinsecticides compiled literature should be provided on the subject.
- "Conclusions" section not good and should be revised. Conclusion should wrap up your review paper with detailed and clear concluding remarks.
Comments on the Quality of English LanguageModerate editing of English language required
Author Response
Answer to the Reviewer II:
Thank you very much for your valuable tips and comments. They have all been considered and revised meticulously. Below are the responses to the Reviewer's comments.
Analyses of this review suggested following revisions:
-Full form for the abbreviation should be provided when they are being used for the first time. E.g EU in abstract
Answer: Corrected as suggested by the Reviewer.
-Botanical name of the Plants should be in Italics.
Answer: The text has been corrected according to the Reviewer's suggestions.
-So many English and typographical mistakes through out the manuscript should be rectified
Answer: The text has been corrected according to the Reviewer's suggestions.
- Authors should provide a complete, structured and systematic summarization on the related key aspects.
Answer: The text has been supplemented according to the Reviewer's suggestions.
-Authors should outline detailed views on future research directions and perspectives.
Answer: The conclusion discusses future research perspectives in the context of the reviewed oilseed plants.
- The numbering of sections and subsections are problematic and must be rectified. A index must be provided for the readers
Answer: Corrected as suggested by the Reviewer.
- Why authors are recommending only two plants for future studies should be clearly explained. What about other plant extracts and essential oils in the literature as bioinsecticides compiled literature should be provided on the subject.
Answer: The literature reviewed suggests that plant extracts and essential oils can negatively affect key insect physiological processes through various mechanisms. The bioactive compounds analysed in the study have been derived from leaves and fruits, but the potential of oil plants such as Camelina sativa and Brassica carinata remains unexplored. These plants are distinctive in that they can thrive in infertile, marginal soils while maintaining a satisfactory oil content in their seeds. As a result, they are commonly used in biofuel production. Although there are already reports on the quality profile of Camelina sativa, studies on the metabolite profile of Brassica carinata seeds have not been conducted so far. Previous analyses have mainly focused on determining the effectiveness of individual compounds, which is time-consuming and costly. A comprehensive approach to the composition of bioactive compounds biosynthesized by plants allows for the use of the extracted group of compounds not only as an insecticide but also as a preparation for deterring insect pests. The volatile metabolite content, mainly terpenes and polyphenols in the oils/extracts from Camelina sativa and Brassica carinata seeds, has enormous biological potential. The use of these components as the main insecticide/repellent in forestry is an innovative approach that has not been used before.
- "Conclusions" section not good and should be revised. Conclusion should wrap up your review paper with detailed and clear concluding remarks.
Answer: The text has been corrected according to the Reviewer's suggestions.
Comments on the Quality of English Language
Moderate editing of English language required
Answer: The text has been corrected according to the Reviewer's suggestions.
Reviewer 3 Report
Comments and Suggestions for Authors
Review Report for Manuscript Forests-2731339
The manuscript, entitled "Oils derived from Camelina sativa and Brassica carinata as potential raw materials in the production of environmentally friendly insecticides” is based on a novel idea and well written. So, I recommend the "minor revision" of this manuscript in current form after handling some minor comments enlisted below.
- Section; Title. The title needs revision, it looks too long try to short it.
- Section; Key words. Arrange key words alphabetically for more clarity.
- Section; References. Cross-check all the enumerated references with the references cited in the text.
- Also check the whole manuscript for some minor English mistakes of grammar and punctuation.
- Also check the whole manuscript for some minor English mistakes of grammar and punctuation.
Author Response
Answer to the Reviewer III:
Thank you very much for your valuable tips and comments. They have all been considered and revised meticulously. Below are the responses to the Reviewer's comments.
So, I recommend the "minor revision" of this manuscript in current form after handling some minor comments enlisted below.
Section; Title. The title needs revision, it looks too long try to short it.
Answer: Corrected as suggested by the Reviewer.
Section; Key words. Arrange key words alphabetically for more clarity.
Answer: Corrected as suggested by the Reviewer.
Section; References. Cross-check all the enumerated references with the references cited in the text.
Answer: Corrected as suggested by the Reviewer.
Also check the whole manuscript for some minor English mistakes of grammar and punctuation.
Answer: Corrected as suggested by the Reviewer.
Comments on the Quality of English Language
Also check the whole manuscript for some minor English mistakes of grammar and punctuation.
Answer: The text has been corrected according to the Reviewer's suggestions.
Reviewer 4 Report
Comments and Suggestions for Authors
The review article entitled "Oils derived from Camelina sativa and Brassica carinata as potential raw materials in the production of environmentally friendly insecticides" needs revision.
1. Please revise your title according to your contents.
2. The authors mentioned the explanation of different groups of insecticides, but they did not mention the effectiveness of these groups against forest insect pests.
3. Please clarify your review aim; your review either focused on forest insect pests or agricultural insect pests.
Please check the annotated pdf file
| . |

N/A
Author Response
Answer to the Reviewer IV:
Thank you very much for your valuable tips and comments. They have all been considered and revised meticulously. Below are the responses to the Reviewer's comments.
- Please revise your title according to your contents.
Answer: Corrected as suggested by the Reviewer.
- The authors mentioned the explanation of different groups of insecticides, but they did not mention the effectiveness of these groups against forest insect pests.
Answer: In general, chemical plant protection compounds are highly effective against forest insects. Therefore, we have focused the reader's attention on the effectiveness of natural substances. However, considering their negative impact on the ecosystem and the residues they leave behind, the authors have decided not to encourage readers by providing exact values. It is also important to note that the use of most chemical insecticides is prohibited in the EU.
- Please clarify your review aim; your review either focused on forest insect pests or agricultural insect pests.
Answer: The review concerned insect pests occurring in forest areas. This has been corrected.
Round 2
Reviewer 1 Report
Comments and Suggestions for Authors
The authors have taken an extensive revision of their work and have cleared all the points which was uploaded in PDF file. It may be accepted now
Comments on the Quality of English LanguageEnglish Language needs minor corrections
Reviewer 2 Report
Comments and Suggestions for Authors
Authors have responded to most of the comments, however, there are still correction required e.g. plant name at various places not in italics. Authors need to be carefully revised the manuscript.
Comments on the Quality of English LanguageMinor editing of English language required
Reviewer 4 Report
Comments and Suggestions for Authors
Well